# Instance-aware 3D Semantic Segmentation powered by Shape Generators and Classifiers

## Abstract

Existing 3D semantic segmentation methods rely on point-wise or voxel-wise feature descriptors to output segmentation predictions. However, these descriptors are often supervised at point or voxel level, leading to segmentation models that can behave poorly at instance-level. In this paper, we proposed a novel instance-aware approach for 3D semantic segmentation. Our method combines several geometry processing tasks supervised at instance-level to promote the consistency of the learned feature representation. Specifically, our methods use shape generators and shape classifiers to perform shape reconstruction and classification tasks for each shape instance. This enforces the feature representation to faithfully encode both structural and local shape information, with an awareness of shape instances. In the experiments, our method significantly outperform existing approaches in 3D semantic segmentation on several public benchmarks, such as Waymo Open Dataset, SemanticKITTI and ScanNetV2.

## 1 Introduction

3D semantic segmentation is fundamental to the perception systems in robotics, autonomous driving, and other fields that require active interaction with the 3D physical surrounding environment. In the deep learning era, a common approach to 3D semantic segmentation is to compute point- or voxel-level descriptors, which are then used to perform point-wise semantic segmentation. So far, most existing approaches have focused on crafting novel neural architectures for descriptor learning and class prediction Zhu et al. (2021); Tang et al. (2020); Choy et al. (2019); Zhang et al. (2020b); Milioto et al. (2019); Xu et al. (2020); Cheng et al. (2022). Few approaches Yan et al. (2021; 2022); Ye et al. (2023) have looked into what type of supervision beyond semantic labels is beneficial for learning dense descriptors for 3D semantic descriptors.

In this paper, we study how instance label supervision can benefit semantic segmentation. Intuitively, in 3D semantic segmentation, the instance labels offer supervision about geometric features of individual objects (e.g., object sizes and most popular shapes) and correlations among those objects. Such supervisions, which are not present from semantic labels, enable learning more powerful descriptors for semantic segmentation. However, compared to obtaining semantic labels, acquiring instance-level labels is more costly, particularly on objects with potentially many instances (e.g., vehicles and pedestrians).

The biggest message of this paper is that in 3D semantic segmentation, instance labels can be computed in an almost unsupervised manner. Moreover, we introduce additional feature learning tasks that are insensitive to erroneous instance labels. Specifically, we introduce a clustering approach that takes an input point cloud, ground-truth semantic labels, and learned dense descriptors from semantic labels and outputs clusters of input points as object instances. The clustering procedure is driven by prior knowledge of the average object size of each object class, which is unique in 3D semantic segmentation compared to 2D semantic segmentation.

Given the predicted instance labels, we introduce two additional tasks that take semantic descriptors as input, i.e., classification and shape reconstruction, to boost descriptor learning. The classification task forces the semantic descriptors to predict instance labels, promoting that semantic descriptors capture instance-level shape features and contextual features. The reconstruction task asks the semantic descriptors to reconstruct the 3D geometry of individual object instances, offering another level of supervision on semantic descriptors to capture instance-level shape features. Note that both

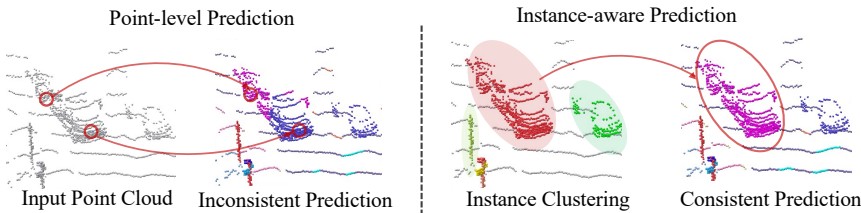

Figure 1: We propose an instance-aware semantic segmentation framework. Baseline methods (left) apply segmentation at the individual point level, without considering the relations between points on the same instance, which causes inconsistent predictions on one object. Our method (right) introduce instance-level classification and reconstruction, making the network learn between instance shape features and getting more consistent and accurate results.

tasks are insensitive to wrong clustering results, making our approach robust with only semantic labels as supervision. In addition, our method is orthogonal to improving network architecture for 3D semantic segmentation and is effective under different feature extraction backbones.

We have evaluated our approach on two outdoor datasets, i.e., SemanticKitti and Waymo, and one indoor dataset, i.e., ScanNetV2. Experimental results show that our approach can boost IoU from the state-of-the-art by 0.7% and 0.9% on SemanticKitti and Waymo, and 0.8% on ScanNetV2. In addition, our approach is competitive against using ground-truth instance labels. For example, on Waymo, using the ground-truth instance labels offers an improvement in IoU by 1.5%. This shows the effectiveness of our clustering approach for identifying individual object instances.

To summarize, our contributions are

- We study instance-level supervision for 3D semantic segmentation and introduce an effective unsupervised approach for identifying object instances.
- Using the predicted object instances, we introduce classification and shape reconstruction as additional tasks to boost semantic descriptor learning.
- We show state-of-the-art results on both indoor and outdoor benchmarks and consistent improvements on variant baselines.

## 2 RELATED WORKS

**3D Semantic Segmentation.**  3D semantic segmentation is fundamental to understanding indoor and outdoor scenes. Current works follow the U-Net structure, where the input point cloud is downsampled and upsampled to obtain per-point features. The resulting point features are then used to predict segmentation labels for each individual points. For indoor scenes Dai et al. (2017); Armeni et al. (2017), the point samples are often uniformly sampled, which is suitable for point-based methods Qi et al. (2016; 2017); Wu et al. (2019); Thomas et al. (2019); Lai et al. (2022); Hu et al. (2020b); Zhao et al. (2021a); Zhang et al. (2020a); Lei et al. (2020); Hu et al. (2020a); Qiu et al. (2021); Yang et al. (2023) Outdoor scenes typically come from LiDAR, point-based methods suffer from efficiency and computation costs due to data sparsity and large data size. People use different data representations to make the U-Net framework more efficient and effective. SqueezeNet Xu et al. (2020), RangeNet++ Milioto et al. (2019), SalsaNext Cortinhal et al. (2020), FIDNet Zhao et al. (2021b), and CENet Cheng et al. (2022) project the input point cloud to a front-view range image and use the 2D convolution networks to do the segmentation. SparseConvNet Graham & van der Maaten (2017), MinkwoskiNet Choy et al. (2019), $(AF)^2S3$Net Cheng et al. (2021) and LidarMultiNet Ye et al. (2023) take advantage of sparse convolution and use volumetric grid to do the 3D segmentation. SPVNAS Tang et al. (2020) combines points and voxel representation to get more accurate results. More novel grid representations are developed to better utilize the LiDAR point cloud properties, such as cylindrical grids Zhu et al. (2021); Hou et al. (2022) and polar BEV coordinates Zhang et al. (2020b). These baselines predict per-point semantic labels separately without considering individual object instances. On the other hand, our method applies instance-level feature grouping and learning, which helps the network learn better features for semantic segmentation.

**3D Multi-task Learning.**  Many approaches combine multiple tasks Liang et al. (2019); Wang et al. (2021); Ye et al. (2023); Zhou et al. (2023); Feng et al. (2021) or different sensor data Yan et al. (2022); Liang et al. (2019); Wang et al. (2021); Yan et al. (2021) to boost the performance of

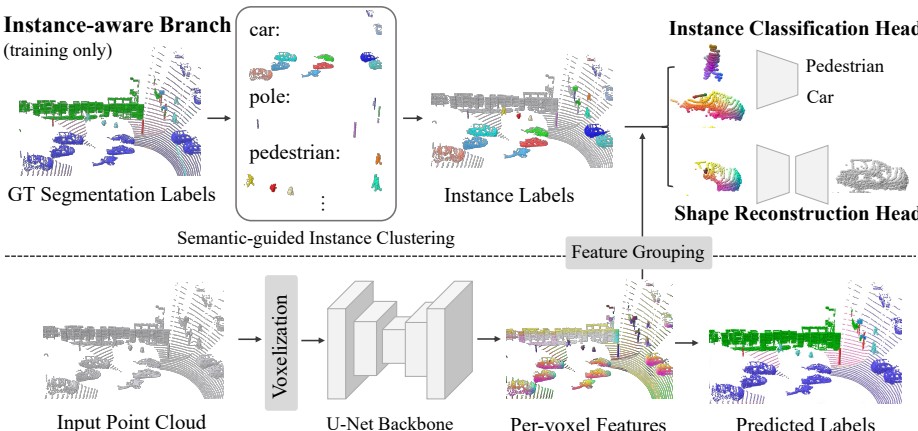

Figure 2: The pipeline of our method. Taking the 3D point cloud as input, the framework outputs the per-point semantic labels. The segmentor is composed of 3D sparse U-Net backbone, and a per-point semantic prediction head. Upon the backbone we add two instance-level branches: instance classification head and shape completion head. Instance labels are obtained by semantic guided clustering. Backbone features are grouped by instance labels and fed into a shape classifier and shape autoencoder. The per-point segmentation, instance classification and shape reconstruction are jointly trained to help the backbone learn better instance-aware features.

single-task learning. LidarMultiNet Ye et al. (2023) combines object detection, BEV segmentation and semantic segmentation. JS3CNet Yan et al. (2021) added semantic scene completion upon the segmentation task. 2DPASS Yan et al. (2022) fuses 2D images with 3D point clouds. All of those papers require more supervision or sensor data, while our method only uses semantic labels and acquires instance labels in an unsupervised way.

**Feature Learning by Completion.** Following Masked Autoencoder (MAE) He et al. (2022a), a lot of methods do the masking and completion-based feature learning and pre-training on 3D shapes Pang et al. (2022); Guo et al. (2023); Yan et al. (2023); Liang et al. (2022); Zhang et al. (2022; 2023) and 3D scenes Min et al. (2022); Hess et al. (2023); Chen et al. (2023). Our method also does feature learning by completion. Unlike those papers, which do the scene or shape level masking and model pre-training for the entire input, our method does the masking on the instance level. In addition, we aim to refine the features of segmentation rather than the autoencoder.

## 3 APPROACH OVERVIEW

In this section, we highlight the design principles of our approach, named with InsSeg, and leave the details to section 4. Figure 2 summarizes our approach. Broadly speaking, our approach adopts a descriptor learning module (introduced in section 4.1) and leverages instance-level supervision tasks without requiring ground-truth instance labels from additional annotation procedures.

Besides the descriptor learning module, our approach uses a semantic-guided instance clustering module and two instance-level supervision heads. The instance clustering module computes instance labels from the input point cloud, ground-truth semantic labels, and the point-wise feature descriptors generated from the descriptor learning module. The instance-level supervision heads perform shape reconstruction and classification to enhance the feature representation at training time.

**Semantic-guided Instance Clustering Module.** Computing instance labels is a challenging task that requires a tolerance of variation in the size and number of instances of each object class. Incorrect instance labels can lead to marginal performance gain from instance-level supervision heads. To this end, we use mean-shift clustering Comaniciu & Meer (2002), which is a modern clustering algorithm that can be efficiently computed and is robust to the number and size of clusters. The combination of point-wise feature descriptors and the input point cloud leads to robust clustering results. In our experiments, the clustering module has a negligible computation cost and can produce reasonable instance labels for multi-task supervision (see Figure 1, right).

**Instance-level Multi-task Supervision.** Given the instance labels computed from the clustering module, we design two supervision tasks that can regularize the feature representation at the in-

stance level, i.e., shape reconstruction and shape classification. These two tasks force the feature representation to encode both categorical and geometric information regarding each object instance, leading to better learned features. Specifically, the shape classification head adopts a max-pool strategy to make the task insensitive to incorrect instance labels. The shape reconstruction head takes descriptors of each masked-out object instance as input and reconstructs the corresponding complete object instance. In the same spirit as MAE He et al. (2022b), the shape reconstruction forces the point-wise descriptors to capture contextual information. The difference in our setting is that contextual information is prioritized at the object level, which is important for semantic segmentation.

## 4 APPROACH

This section presents the technical details of our approach. We begin with the descriptor learning backbone in Section 4.1. We then introduce the clustering module in Section 4.2. Section 4.3 and Section 4.4 introduce the classification and reconstruction heads, respectively. Finally, Section 4.5 elaborates on the technical details.

### 4.1 DESCRIPTOR LEARNING

As shown in Figure 2, the descriptor learning module combines a voxelization sub-module and a 3D U-Net sub-module for descriptor extraction. The voxelization sub-module transforms the point cloud $\mathcal{P} \in R^{N \times 3}$ into a fix-sized voxel grid and extracts initial voxel features $\mathcal{F}_0 \in R^{M \times d_0}$ by aggregating points in same voxels, where $M$ is the number of non-empty voxels. The 3D U-Net sub-module then takes $\mathcal{F}_0$ as input and compute backbone features $\mathcal{F} \in R^{M \times d}$ by a multi-scale encoder-decoder model. The backbone features are fed into the voxel semantic prediction head $H^s$ and per-voxel semantic labels $\bar{\mathcal{S}}^V \in R^M$ are predicted.

### 4.2 INSTANCE LABELS VIA CLUSTERING

The clustering module takes the predicted point-wise descriptors (obtained by interpolating the output of the 3D U-Net), ground-truth semantic labels, and 3D point coordinates as input and outputs clusters of points, each of which corresponds to one predicted object instance. Our major goal is to leverage prior knowledge on the average shape size of each object class to identify object instances.

Specifically, we perform mean-shift clustering Comaniciu & Meer (2002) on points of each scene that belong to the same semantic class. The feature vector $f = (p, \lambda_p, d)$ for each point $p$ includes its location $p$ and its descriptor returned by the descriptor learning module. $\lambda_p = 1/\sigma_p$ where $\sigma_p$ is the descriptor variance among all points whose distance to $p$ are within $r_p$ where $r_p$ is based on the semantic class label of $p$, e.g., $r_p = 1$m for cars and $r_p = 0.5$m for pedestrians. Please refer to the supplementary material for details and the visualization of instance clustering.

We denote the resulting clusters as $\{O_k | k = 1, ..., K\}$. Each $O_k$ corresponds to one object semantic label $c_k$. The raw point cloud and backbone features in instance $O_k$ are denoted by $\mathcal{P}(O_k) \in R^{M_k \times 3}$ and $\mathcal{F}(O_k) \in R^{M_k \times d}$, respectively, where $M_k$ is the number of voxels falling on instance $O_k$ and $d$ is the feature dimension. We also denote the group of voxel centers that falls into instance $O_k$ as $\mathcal{V}(O_k) \in R^{M_k \times 3}$.

Based on the predicted object labels, we design two prediction heads that take the voxel-based semantic descriptors as input. In the same spirit as multi-task learning, our goal is to use these prediction heads to boost the quality of the voxel-based semantic descriptors, which then improve semantic segmentation performance. Both tasks are defined so that they are insensitive to potentially noisy object labels.

### 4.3 INSTANCE CLASSIFICATION HEAD

The first additional prediction head is the instance classification head $H^c$. The goal is to help the network to learn instance-wise semantic features. It groups the backbone features falling in different instances and predicts the semantic categories $\bar{\mathcal{C}}$ for those instances.

For instance $O_k$, we have the grouped backbone features $\mathcal{F}(O_k)$. We first use a max-pooling to aggregate the input features from the voxel-level to instance level, then apply a classification head on this pooled feature. The max-pooling operator can accommodate objects with different number of points and tolerate the errors in the instance clustering, e.g. the feature aggregation of two objects is still valid for single-object classification. With this setup, the predicted class label for the object $O_k$ is

$$\bar{c}_k = \text{MLP}(\text{max-pool}(\mathcal{F}(O_k))), \tag{1}$$

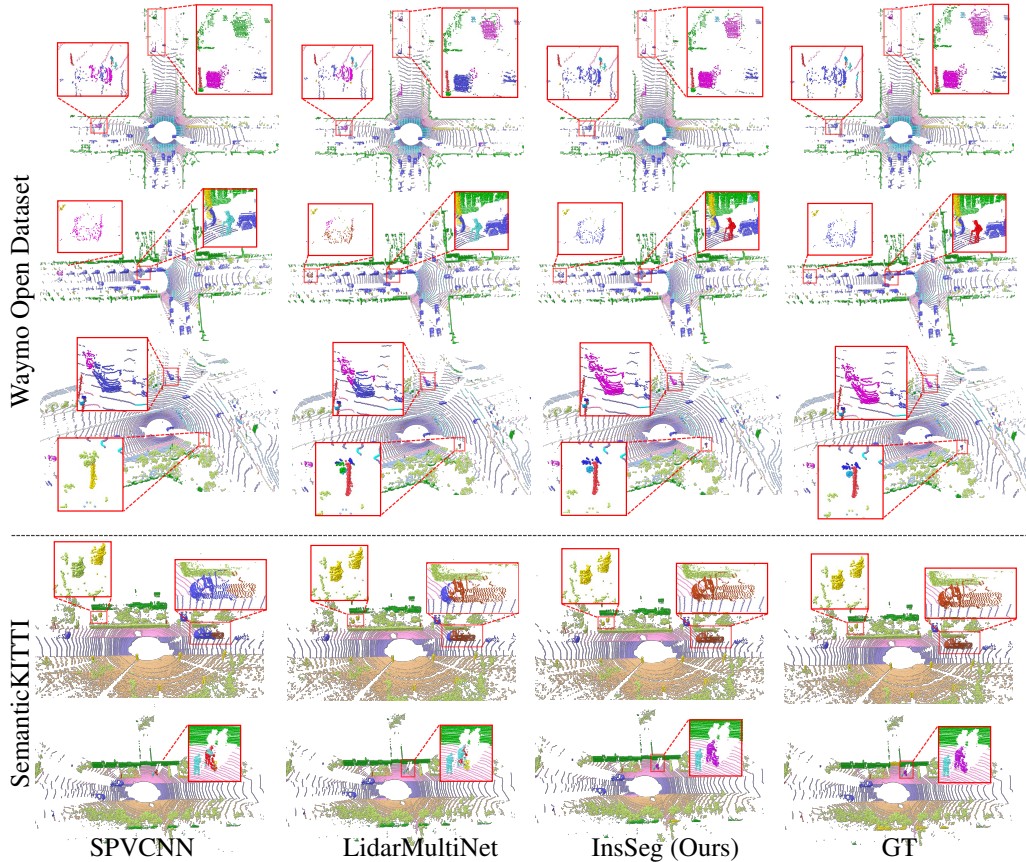

Figure 3: Qualitative results on Waymo Open Dataset and SemanticKTTI validation set. From left to right we show the semantic segmentation results of SPVCNNTang et al. (2020), LidarMultiNetYe et al. (2023), our method, and the ground truth semantic labels. We use red boxes to highlight the inconsistent or erronous predictions from baselines. Our method is able to improve the consistency and accuracy of the semantic prediction on objects.

The loss function adopted for this head is the instance classification loss between the predicted object class labels $\bar{c}_k$ and the ground truth labels $c_k$. We use OHEM lossShrivastava et al. (2016) in our method:

$$L^c = \frac{1}{K} \sum_{k=0}^{K} l_{\text{ohem}}(\bar{c}_k, c_k) \tag{2}$$

### 4.4 SHAPE RECONSTRUCTION HEAD

The second prediction head is $H^g$, which takes backbone features from part of an instance (a subset of a full instance) and aims to reconstruct the geometry of the full instance. This prediction head takes motivations from MAE He et al. (2022a), which performs feature learning by completing masked out regions. Our approach presents two fundamental differences. First, the input are point descriptors not raw 3D points. The point descriptors are jointly trained by semantic labels, providing good initializations for feature learning. Second, in contrast to reconstructing the entire scene, the reconstruction is performed at the instance level. This helps the features capture object-specific features for semantic segmentation.

Specifically, to get the input for the shape reconstruction head of instance $O_k$, we randomly mask part of the backbone feature $\mathcal{F}(O_k)$. This is done by randomly choosing a voxel $q$ from the voxel centers $\mathcal{V}(O_k)$ and mask all voxels within radius $r$ (which depends on the semantic class and is the same as the one used for clustering) from $q$. The input for the completion head is

$$\mathcal{F}'(O_k) = \mathcal{F}(O_k)[\text{mask}(q, r)] \tag{3}$$

| | mIoU | car | bicycle | motorcycle | truck | other vehicle | person | bicyclist | motorcyclist | road | parking | sidewalk | other ground | building | fence | vegetation | trunk | terrain | pole | traffic sign |
|---|---|---|---|---|---|---|---|---|---|---|---|---|---|---|---|---|---|---|---|---|
| RangeNet++Milioto et al. (2019) | 52.2 | 91.4 | 25.7 | 34.4 | 25.7 | 23.0 | 38.3 | 38.8 | 4.8 | 91.8 | 65.0 | 75.2 | 27.8 | 87.4 | 58.6 | 80.5 | 55.1 | 64.6 | 47.9 | 55.9 |
| PolarNetZhang et al. (2020b) | 54.3 | 93.8 | 40.3 | 30.1 | 22.9 | 28.5 | 43.2 | 40.2 | 5.6 | 90.8 | 61.7 | 74.4 | 21.7 | 90.0 | 61.3 | 84.0 | 65.5 | 67.8 | 51.8 | 57.5 |
| SqueezeSegV3Xu et al. (2020) | 55.9 | 92.5 | 38.7 | 36.5 | 29.6 | 33.0 | 45.6 | 46.2 | 20.1 | 91.7 | 63.4 | 74.8 | 26.4 | 89.0 | 59.4 | 82.0 | 58.7 | 65.4 | 49.6 | 58.9 |
| KPConvThomas et al. (2019) | 58.8 | 95.0 | 30.2 | 42.5 | 33.4 | 44.3 | 61.5 | 61.6 | 11.8 | 90.3 | 61.3 | 72.7 | 31.5 | 90.5 | 64.2 | 84.8 | 69.2 | 69.1 | 56.4 | 47.4 |
| SalsaNextCortinhal et al. (2020) | 59.5 | 91.9 | 48.3 | 38.6 | 38.9 | 31.9 | 60.2 | 59.0 | 19.4 | 91.7 | 63.7 | 75.8 | 29.1 | 90.2 | 64.2 | 81.8 | 63.6 | 66.5 | 54.3 | 62.1 |
| FIDNetZhao et al. (2021b) | 59.5 | 93.9 | 54.7 | 48.9 | 27.6 | 23.9 | 62.3 | 59.8 | 23.7 | 90.6 | 59.1 | 75.8 | 26.7 | 88.9 | 60.5 | 84.5 | 64.4 | 69.0 | 53.3 | 62.8 |
| BAAFQiu et al. (2021) | 59.9 | 95.4 | 31.8 | 35.5 | 48.7 | 46.7 | 49.5 | 55.7 | 53.0 | 90.9 | 62.2 | 74.4 | 23.6 | 89.8 | 60.8 | 82.7 | 63.4 | 67.9 | 53.7 | 52.0 |
| CENetCheng et al. (2022) | 59.4 | 91.7 | 43.0 | 40.5 | 42.3 | 42.8 | 55.1 | 58.4 | 26.3 | 90.5 | 65.6 | 74.0 | 30.4 | 89.1 | 61.4 | 81.4 | 60.4 | 66.2 | 50.4 | 59.3 |
| FusionNetZhang et al. (2020a) | 61.3 | 95.3 | 47.5 | 37.7 | 41.8 | 34.5 | 59.5 | 56.8 | 11.9 | 91.8 | 68.8 | 77.1 | 30.8 | 92.5 | 69.4 | 84.5 | 69.8 | 68.5 | 60.4 | 66.5 |
| Cylinder3D Zhu et al. (2021) | 61.8 | 96.1 | 54.2 | 47.6 | 38.6 | 45.0 | 65.1 | 63.5 | 13.6 | 91.2 | 62.2 | 75.2 | 18.7 | 89.6 | 61.6 | 85.4 | 69.7 | 69.3 | 62.6 | 64.7 |
| SPVCNNTang et al. (2020) | 63.6 | 96.2 | 49.4 | 47.1 | 45.2 | 40.6 | 60.8 | 68.3 | 41.1 | 90.9 | 63.9 | 75.4 | 17.9 | 91.3 | 66.5 | 85.5 | 71.1 | 69.6 | 61.6 | 65.4 |
| MinkowskiNetChoy et al. (2019) | 64.3 | 96.2 | 43.7 | 46.4 | 51.6 | 41.0 | 62.8 | 68.1 | 49.6 | 90.9 | 64.0 | 75.2 | 21.7 | 90.5 | 64.5 | 85.6 | 70.2 | **69.8** | 60.8 | **66.5** |
| LidarMultiNetYe et al. (2023) | 64.3 | 96.2 | 45.7 | 47.0 | 47.1 | 40.9 | 62.7 | 66.8 | 40.8 | 90.7 | 67.0 | 75.5 | 29.3 | 92.0 | 67.5 | 85.3 | **72.5** | 69.7 | 61.5 | 62.5 |
| InsSeg | **65.0** | **96.5** | 37.5 | **48.9** | **51.1** | 44.1 | **64.3** | **69.0** | 42.5 | 91.2 | 66.9 | 76.1 | 30.8 | **92.5** | 69.2 | **85.7** | 72.3 | **69.8** | **61.7** | 64.1 |

Table 1: Quantitative semantic segmentation results on SemanticKITTIBehley et al. (2019) test set. We show the IoU for 23 semantic classes and mIoU among them. Our method outperforms all baselines in mIoU and most categories.

| | mIoU | car | truck | bus | other vehicle | motorcyclist | bicyclist | pedestrian | sign | traffic light | pole | cone | bicycle | motorcycle | building | vegetation | tree trunk | curb | road | lane marker | other ground | walkable | sidewalk |
|---|---|---|---|---|---|---|---|---|---|---|---|---|---|---|---|---|---|---|---|---|---|---|---|
| RPVNetXu et al. (2021) | 62.6 | 94.8 | 67.4 | 74.9 | 33.0 | 0.0 | 77.3 | 88.6 | 68.0 | 28.6 | 74.7 | 37.6 | 53.8 | 64.9 | 96.5 | 86.4 | 67.1 | 70.9 | 90.9 | 23.7 | 24.0 | 68.8 | 84.4 |
| SPVCNNTang et al. (2020) | 64.12 | 94.68 | 68.82 | 73.59 | 26.89 | 0.04 | 73.77 | 88.08 | 68.50 | 26.75 | 74.30 | 43.31 | 56.78 | 62.34 | 96.32 | 85.93 | 66.58 | 70.78 | 91.90 | 41.37 | 46.66 | 69.69 | 83.67 |
| Cylinder3DZhu et al. (2021) | 64.19 | 94.21 | 66.75 | 76.76 | 25.85 | 0.10 | 74.27 | 88.01 | 65.61 | 26.64 | 73.92 | **49.59** | 58.07 | 66.12 | 96.51 | 86.28 | 67.77 | 66.22 | 91.42 | 37.26 | 45.23 | 71.75 | 84.86 |
| MinkowskiNetChoy et al. (2019) | 65.06 | 95.06 | 69.48 | 77.27 | 29.52 | 0.00 | 74.34 | 88.40 | **68.98** | 28.53 | 75.92 | 48.67 | 57.58 | 64.62 | 96.47 | 86.26 | 67.98 | 71.32 | 92.05 | 41.46 | 45.09 | 70.39 | 84.08 |
| LidarMultiNetYe et al. (2023) | 65.24 | 94.42 | 65.70 | 76.37 | 29.47 | 0.05 | **78.07** | 89.57 | 68.34 | 28.55 | **76.02** | 48.35 | 57.79 | 65.85 | **96.70** | 86.87 | 67.93 | 72.22 | **92.41** | 45.02 | 48.17 | 71.25 | 84.85 |
| InsSeg | **66.13** | **95.09** | **69.78** | **79.83** | **30.13** | **0.11** | 77.38 | **89.59** | 68.25 | **28.77** | 75.67 | 48.19 | **58.42** | **66.66** | 96.68 | **87.09** | **68.44** | **72.25** | 92.38 | **45.05** | **48.27** | **71.80** | **84.93** |

Table 2: Quantitative semantic segmentation results on Waymo Open DatasetSun et al. (2020) test set. Our method shows superior results to baselines.

where mask$(q, r)_i$=True if $\|\mathcal{V}(O_k)_i - q\| > r$, else False, for $i = 1, ..., M_k$. The dimension of $\mathcal{F}'(O_k)$ is $M_k' \times d$, and $M_k' < M_k$. The reconstruction head $H^g$ then takes the masked backbone features $\mathcal{F}'(O_k)$ as input, and output the raw voxel center locations $\mathcal{V}(O_k)$.

We use PointNet AutoencoderQi et al. (2016) as the model architecture. We normalize the voxel locations for each instance with zero mean and pad the voxel number to a same number $N^g$. We use chamfer distance (CD) as the objective function:

$$L^g = CD\big(H^g(\mathcal{F}'(O_k), \mathcal{V}(O_k)\big) \tag{4}$$

### 4.5 Training Details

We perform training in two stages. In the first stage, we drop the classification and reconstruction heads and train the descriptor learning module and the semantic segmentation head using semantic labels. We then use the resulting descriptors to predict object instances. After that, we activate classification and reconstruction heads and train all the modules together. The total loss term is

$$L = L^s + \lambda_1 L^c + \lambda_2 L^g \tag{5}$$

where $L^s$, $L^c$ and $L^g$ are per-point segmentation loss, instance classification loss, and shape reconstruction loss, respectively. $\lambda_1$ and $\lambda_2$ are weights of different loss terms. In our method we set $\lambda_1 = 0.1$ and $\lambda_2 = 0.01$. We train our model on 8 Tesla V100 GPUs with batch size 2 for 30 epochs. The first 10 epochs are for descriptor learning and last 20 epochs are for joint training with instance supervision heads. We use Adam optimizer and OneCycleLRSmith & Topin (2018) scheduler with the starting learning rate 0.003 for outdoor datasets Sun et al. (2020); Behley et al. (2019) and 0.03 for ScanNetDai et al. (2017).

## 5 Experiments

### 5.1 Outdoor Scene Semantic Segmentation

**Datasets**. We conduct our experiments on two large-scale LiDAR datasets: SemanticKITTI DatasetBehley et al. (2019) and Waymo Open Dataset Sun et al. (2020). SemanticKITTI contains 22 driving sequences, where sequences 00-07, 09-10 are used for training, 08 for validation, and 11-21 for testing. A total number of 19 semantic classes are chosen following the SemanticKITTI benchmark. For Waymo Open Dataset, it contains 1150 sequences in total, where 798 sequences are used for training, 202 for validation, and 150 for testing. Each sequence contains about 200 frames of LiDAR point cloud. For the semantic segmentation task, there are 23,691 and 5,976 frames with

semantic segmentation labels in the training and validation set, respectively. There are a total of 2,982 frames in the test set. 23 semantic classes are chosen following the WOD benchmark.

**Instance Choices**. We manually choose thing objects in both datasets. Apart from foreground objects like vehicles and pedestrians, we also include background objects such as poles, traffic lights. Specifically, for SemanticKITTI, we choose 12 instance classes including car, bicycle, motorcycle, truck, other vehicle, person, bicylist, motorcyclist, trunk, pole and traffic sign. For Waymo Open Dataset, we choose 13 classes: car, truck, bus, other vehicle, motorcyclist, bicyclist, pedestrian, sign, traffic light, pole, cone, bicycle, and motorcycle.

**Evaluation Metric**. We adopt the intersection-over-union (IoU) of each class and the mean IoU (mIoU) of all classes as the evaluation metric. The IoU for class $i$ is $\text{IoU}_i = \frac{TP_i}{TP_i+FP_i+FN_i}$, where $TP_i, FP_i, FN_i$ are true positive, false positive, and false negatives for class $i$. The mean IoU (mIoU) is calculated by $\text{mIoU} = \frac{1}{N}\sum_{i=1}^{N}\text{IoU}_i$. Apart from this, we also adopt an instance-level metric called instance classfication accuracy. The details can be found in Section 5.1.2.

**Baseline Methods**. We compare our method with state-of-the-art semantic segmentation methodsYe et al. (2023); Zhu et al. (2021); Tang et al. (2020); Cheng et al. (2022); Cortinhal et al. (2020); Zhang et al. (2020a); Milioto et al. (2019); Zhang et al. (2020b); Xu et al. (2020); Thomas et al. (2019); Zhao et al. (2021b); Qiu et al. (2021); Choy et al. (2019). Methods with extra input information (e.g. 2D image) or extra supervision (e.g. object detection and semantic scene completion) are not included in the comparison. For papers without code releasingYe et al. (2023), we implemented their methods according to their papers and include the coding details in the supplementary materials.

### 5.1.1 RESULTS ON SEMANTICKITTI AND WAYMO OPEN DATASET

In this section we show the outdoor LiDAR semantic segmentation on SemanticKITTI DatasetBehley et al. (2019) and Waymo Open DatasetSun et al. (2020). Table 1 and Table 2 show the mIoU and per-class IoU on the test set of both datasets. Our method achieves the state-of-the-art results compared with various baselines. Figure 3 shows the qualitative comparison of our method with baselines. Zoom-in box highlights some inconsistent or wrong predictions of the baseline methods. Our method is able to get the consistent and accurate segmentation results, both on foreground and background objects.

### 5.1.2 INSTANCE CLASSIFICATION ACCURACY METRIC

Our method combines the point-based and instance-based information in the semantic segmentation. The mIoU is not enough to show the advantage of our method because it only computes per-point accuracies. Here we proposed a instance-level metric: classification accuracy for segmentation $\text{Acc}_{\text{seg}}$. It computes the ratio of the correctly classified objects in all objects with the same semantic label. For the semantic label $c$, the classification accuracy is defined as $\text{Acc}_{\text{seg}}^c = \frac{N_{\text{correct}}^c}{N^c}$, where $N_{\text{correct}}$ and $N_c$ are the correctly classified and total number of instances with semantic label $c$.

For an object $O_k$ with $m_k$ points and semantic label $c$, we say it's correctly classifier if the ratio of correctly predicted points is above some threshold $t$: $\frac{N_{\text{pred}_k=c}}{m_k} \geq t$.

In the Table 3 we show the results on the 7 foreground categories which has the ground-truth instance labels. We show results with two different thresholds $0.5$ and $0.8$. We have significantly improvement on the classes where the inconsistent classification often happens, e.g. 12% on the bus and 25% on the other vehicle.

| | Mean Acc | | car | | truck | | bus | | other vehicle | | motorcyclist | | bicyclist | | pedestrian | |
|---|---|---|---|---|---|---|---|---|---|---|---|---|---|---|---|---|
| Threshold | 0.5 | 0.8 | 0.5 | 0.8 | 0.5 | 0.8 | 0.5 | 0.8 | 0.5 | 0.8 | 0.5 | 0.8 | 0.5 | 0.8 | 0.5 | 0.8 |
| LidarMultiNetYe et al. (2023) | 0.637 | 0.594 | 0.929 | 0.913 | 0.575 | 0.525 | 0.641 | 0.576 | 0.294 | 0.195 | 0.337 | 0.315 | 0.823 | 0.798 | 0.85.9 | 0.833 |
| InsSeg | 0.654 | 0.614 | 0.931 | 0.912 | 0.584 | 0.544 | 0.700 | 0.645 | 0.334 | 0.244 | 0.315 | 0.293 | 0.843 | 0.815 | 0.868 | 0.842 |

Table 3: Instance classification accuracy under different thresholds on the Waymo validation set.

### 5.2 INDOOR SCENE SEMANTIC SEGMENTATION

**Dataset**. ScanNetDai et al. (2017) is an RGB-D video dataset collected from indoor environments. The training/validation/test set includes 1201, 312, 100 scans respectively. The dataset provides labels for 40 common indoor semantic classes, while only 20 of them are used for performance evaluation. We use 21 classes in training with class 0 to be not evaluated classes.

**Instance Choices**. We choose all instances categories except wall and floor.

**Metric**. We use the same metric IoU as the outdoor dataset.

| | mIoU | bathtub | bed | bookshelf | cabinet | chair | counter | curtain | desk | door | floor | other furni. | picture | refrigerator | shwr curtain | sink | sofa | table | toilet | wall | window |
|---|---|---|---|---|---|---|---|---|---|---|---|---|---|---|---|---|---|---|---|---|---|
| PointNet++ Qi et al. (2017) | 62.62 | 81.47 | 74.22 | 68.93 | 53.60 | 85.33 | 48.56 | 61.31 | 51.89 | 45.19 | 94.02 | 40.57 | 25.55 | 46.29 | 53.02 | 63.19 | 73.44 | 65.75 | 86.74 | 78.81 | 54.61 |
| PointNet++ + InsSeg | 63.46 | 82.34 | 75.83 | 69.01 | 55.21 | 85.98 | 53.25 | 63.57 | 54.42 | 44.21 | 93.86 | 41.23 | 28.56 | 46.21 | 57.35 | 61.21 | 75.33 | 66.75 | 86.41 | 76.32 | 52.11 |
| MinkwosikiNet Choy et al. (2019) | 67.35 | 84.60 | 76.30 | 73.74 | 59.84 | 88.75 | 60.93 | 68.12 | 56.57 | 55.89 | 94.53 | 47.13 | 18.16 | 56.35 | 62.62 | 70.33 | 75.90 | 68.44 | 91.88 | 81.81 | 55.03 |
| MinkwosikiNet + InsSeg | 68.18 | 86.12 | 78.02 | 72.17 | 61.22 | 88.86 | 62.23 | 70.04 | 59.81 | 55.58 | 94.52 | 47.77 | 20.54 | 51.86 | 68.86 | 69.23 | 79.44 | 70.59 | 90.81 | 81.77 | 54.14 |
| Stratified Lai et al. (2022) | 74.05 | 89.47 | 81.44 | 81.20 | 66.00 | 89.83 | 66.56 | 73.57 | 71.53 | 70.34 | 95.86 | 55.83 | 32.04 | 64.35 | 67.22 | 65.56 | 83.78 | 76.53 | 94.45 | 86.67 | 68.85 |
| Stratified + InsSeg | 74.47 | **91.71** | 81.67 | 81.79 | 64.93 | 90.07 | 66.78 | 71.45 | **73.25** | **71.36** | 95.47 | 54.37 | 34.23 | 64.89 | 71.68 | 65.53 | **85.52** | 73.42 | **95.14** | 86.88 | 69.06 |
| Swin3D Yang et al. (2023) | 75.26 | 88.12 | 83.53 | **84.23** | 65.65 | 90.38 | 66.30 | 79.45 | 67.91 | 69.03 | 96.27 | 59.90 | 42.25 | 70.89 | 71.02 | 61.19 | 80.72 | 77.23 | 93.57 | **87.57** | 70.02 |
| Swin3D + InsSeg | **75.87** | 89.23 | 83.21 | 84.11 | **66.34** | **90.83** | **67.04** | **80.39** | 70.88 | 68.53 | **96.33** | **60.56** | **43.39** | **71.16** | **73.82** | 59.78 | 81.92 | **78.01** | 93.32 | 87.41 | **71.15** |

Table 4: Quantitative semantic segmentation results on ScanNet Dai et al. (2017) validation set. We show the segmentation results of baselines with and without our method. The results shows oue method is able to improve upon variant baselines.

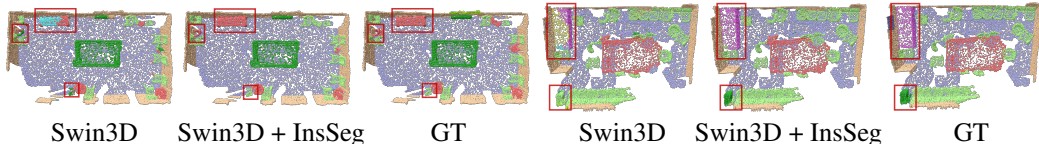

Swin3D    Swin3D + InsSeg    GT         Swin3D    Swin3D + InsSeg    GT

Figure 4: Qualitative results on ScanNet validation set Dai et al. (2017). We show two groups of results of Swin3D Yang et al. (2023) with and without our instance heads. Our method improves the prediction accuracy and object-level consistency.

**Baseline Methods**. We compare our method with the classic and state-of-the-art indoor scene segmentation methods Lai et al. (2022); Choy et al. (2019); Qi et al. (2017); Yang et al. (2023). Here we directly add the instance multi-task learning on top of the baseline methods and do the results comparison.

### 5.2.1 RESULTS ON SCANNET DATASET

Different from the sparse outdoor datasets, indoor scenes are often more dense and have smaller ranges. We follow the majority baselines and use the point-based method. Since most baselines follow the similar high-level structures, backbone and per-point segmentation head, it's easy for us to apply our method on top of their backbones. Here we conduct our experiment by adding the instance clustering and multi-task heads on various baselines. Table 4 and Figure 4 show the quantitative and qualitative comparison of our method with the baseline methods. Table 4 shows that our method improves the segmentation results over various baselines. Figure 4 shows the instance heads improves the prediction consistency significantly.

### 5.3 ABLATION STUDY

In this section, we show the ablation study of our method. Section 5.3.1 shows the effect of our method training on different object categories. Section 5.3.2 shows our method working on different representations of point cloud. Section 5.3.3 shows the results with clustered and GT instance labels. Section 5.3.4 analyzes the effect of different components of our method.

### 5.3.1 CHOICES OF INSTANCE CATEGORIES

As we introduced in Section 5.1, we manually choose 13 object categories on Waymo Open Dataset and 12 classes on SemanticKITTI Dataset. Those categories include vehicles, motorcyclists, street signs, etc. Here we try only train the model on one or several categories and see the improvements on those categories. Table 5a shows the results training with vehicles and motorcyclists instances on Waymo Open Dataset. We get significant improvement on training categories. Note that by training on car, truck, bus, and other vehicle, our method has significant improvement on the last 3 classes, which shows our method is able to improve more on minor classes. This is also consistent with the visual results in Figure 3.

### 5.3.2 BACKBONES WITH DIFFERENT REPRESENTATIONS

In this section, we show that our method not only generalize among different datasets, it also has improvements on various types of 3D representations. To apply our method on different baselines, we keep the backbone and segmentation head and add the instance classification head and completion head upon the per-point or per-voxel backbone features. Table 6a shows the results of our method with point, voxel, point + voxel, and cylinder based representations. Our method is able to

|  | mIoU | Motorcyclist | Bicyclist | Motorcycle | Bicycle |
|---|---|---|---|---|---|
| Baseline | 68.03 | 0.50 | 64.39 | 67.87 | 70.36 |
| Baseline + Motorcyclist | 68.22 | 2.08 | 69.69 | 68.64 | 67.51 |

|  | mIoU | Car | Truck | Bus | Other-veh |
|---|---|---|---|---|---|
| Baseline | 68.03 | 95.26 | 64.92 | 84.69 | 35.53 |
| Baseline + vehicle | 68.53 | 95.06 | 65.03 | 84.86 | 39.78 |

(a) Results of instance heads training on the motorcyclist (top) and four vehicle classes (bottom). Our method improves significantly on classes that are easy to be mis-classified by baselines.

| Baseline | Classification | Reconstruction | mIoU |
|---|---|---|---|
| ✓ |  |  | 68.03 |
| ✓ | ✓ |  | 68.65 |
| ✓ |  | ✓ | 68.48 |
| ✓ | ✓ | ✓ | 68.93 |

(b) Ablation study on different components of our method. Both the instance classification head and the shape reconstruction head contribute to the final results.

Table 5: Ablation study on object categories and different components. Both results are the mIoU on the validation set of Waymo Open DatasetSun et al. (2020).

| Method | PointNet++ | Cylinder3D | SPVCNN | LidarMultiNet |
|---|---|---|---|---|
| Backbone Type | Point | Cylinder | Point + Voxel | Voxel |
| w/o Instance Heads | 64.62 | 65.71 | 66.36 | 68.22 |
| w. Instance Heads | 64.90 | 66.25 | 67.17 | 68.79 |

(a) Generalization of our method on different 3D representations. Our method shows consistent improvement on all of those baselines with different representations.

|  | Clustered Instance Label | GT Instance Label |
|---|---|---|
| mIoU | 68.63 | 69.01 |

(b) Results of InsSeg training on foreground instance categories with clustered instance labels and ground truth instance labels. Note both experiments only use 7 classes with GT instance labels.

Table 6: Ablation study on different 3D representations and instance label sources. All results are on the validation set of Waymo Open Dataset Sun et al. (2020).

get improvements on all of them, which shows our method is a general framework that can be easily inserted into those baselines, with almost no extra computation cost.

### 5.3.3 RESULTS WITH THE GT INSTANCE LABEL

We use the semantic-guided instance clustering to get the instance labels. To show the effectiveness of this method, we show the comparison between our method with the clustered instance labels and ground-truth instance labels in Table 6b on the validation set of Waymo Open Dataset Sun et al. (2020). Note that in this dataset instance labels are only available for 7 foreground classes while our method is able to get labels for 13 classes. To keep the fairness, we conduct both experiments on 7 foreground categories. The results show that our method is robust to the unsupervised instance labels and there's a small margin between the clustered and ground-truth instance labels.

### 5.3.4 ANALYSIS ON DIFFERENT COMPONENTS OF OUR METHOD

We analyze the effectiveness of the instance classification head and the shape reconstruction head by removing the corresponding head and evaluate the results. Table 5b shows the results on the validation set of Waymo Open DatasetSun et al. (2020) with removing different heads. Both instance heads contribute to the final results, where the classification head helps the backbone features capture more shape global features while the reconstruction head preserves more local geometries.

## 6 CONCLUSION, LIMITATIONS AND FUTURE WORK

We proposed InsSeg, an instance-level multi-task framework for 3D semantic segmentation. With instance labels obtained from unsupervised semantic guided clustering, we add two novel branches upon the U-Net style segmentation backbones: instance classification and shape reconstruction. The network learns better shape features with these instance supervision heads and produces consistent predictions on the same objects. Our method achieved state-of-the-art segmentation results on both indoor and outdoor datasets. Moreover, it can generalize to most backbones and improve the results with almost no extra computation burden.

**Limitations**. Our method has two limitations. First, it relies on the property that 3D objects are easy to isolate, so instance clustering can work well. For 2D images, where objects don't have clear boundaries, an unsupervised instance label is hard to get, and our method would fail. Second, since our features are more consistent on the shape level if the instance classification is wrong, the whole object will be mis-segmented, which causes lower IoU than the inconsistent predictions. For failure cases, please refer to the supplementary materials.

**Future Work**. In the future, how to combine point- or voxel-level supervision with more high-level concepts will be a good research topic. Apart from the instance categories and geometry shapes studied in this paper, we can also use object interactions, scene graph priors, and more geometric primitives. How to deploy our method in 2D is also an open area, especially how to get unsupervised instance labels.

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
