# OpenReview forum: "Instance-aware 3D Semantic Segmentation powered by Shape Reconstruction and Classification"
_ICLR.cc/2024/Conference — ICLR 2024 Conference Withdrawn Submission_

### Official Review · Reviewer_9ujj · 2023-10-27

**Soundness:** 4 excellent
**Presentation:** 3 good
**Contribution:** 2 fair
**Rating:** 3
**Confidence:** 4

**Summary:**

The author introduces InsSeg, a 3D semantic segmentation model with an instance-level approach. The author incorporates classification and shape rebuilding tasks to enhance the learning of semantic descriptors, leveraging the forecasted object instances. Experiment results show the effectiveness of the method.

**Strengths:**

The writing of the article is easy to understand. The author has provided detailed methodological and experimental details, leading me to believe that the methods presented in this paper are replicable.

The method proposed in this paper has achieved state-of-the-art results.

**Weaknesses:**

The paper adopts the setup from 3D instance semantic segmentation to address the 3D semantic segmentation task, a practice that lacks innovation.

I am not convinced that the instance labels obtained through clustering in this paper are highly accurate, and inaccurate labels can negatively impact the performance of the classification task designed subsequently. In fact, we can observe from the test set of SemanticKITTI Behley that the method proposed in this paper performs poorly across many categories, suggesting that this approach may not possess strong generalization capabilities.

**Questions:**

see weakness

---

### Official Review · Reviewer_fWVW · 2023-10-29

**Soundness:** 1 poor
**Presentation:** 2 fair
**Contribution:** 2 fair
**Rating:** 3
**Confidence:** 4

**Summary:**

This paper proposes a method for the semantic segmentation of 3D point clouds. The key idea of the proposed method is to leverage instance information, i.e., classification through instance segmentation, for semantic segmentation tasks. The instance information is extracted from the semantic segmentation datasets using a clustering-based method. The experiment shows that the proposed method outperforms several baseline methods.

**Strengths:**

+ Incorporating instance information into a semantic segmentation task would be a reasonable idea. While the panoptic segmentation tasks may implicitly leverage this information, I guess this work is the first to directly incorporate instance information into a semantic segmentation task. If the fundamental idea is new, these discussions would be valuable to the community.

**Weaknesses:**

### Related work
The paper does not discuss existing knowledge about instance segmentation in any part of the paper. This is quite unnatural to me; the key to the proposed method is leveraging what instance segmentation does to the semantic segmentation task. Since the relation of the proposed method to instance and segmentation methods is not described clearly, it is difficult to assess the value of this study correctly.


### Experiment
The experiment seems problematic. The paper uses SemanticKITTI and Waymo Open Dataset for comparison, and both experiments have potential issues as follows:

SemanticKITTI:
The paper only compares old baselines (before 2021) except to LidarMultiNet (Yu et al., AAAI2023). However, Yu et al. did not show an experiment for SemanticKITTI in their original paper (this may be natural since this method was developed for the Waymo competition), and thus, it is unclear if this method achieves SOTA performance to this dataset.
Actually, through the recent leaderboard, we see that several recent methods (after 2022, without using extra training data) dominantly outperform (test mIoU > 70) the proposed method (65.0).
https://paperswithcode.com/sota/3d-semantic-segmentation-on-semantickitti

Waymo Open Dataset:
From the original paper of LidarMultiNet (Yu et al., AAAI2023), we see the authors fail to reproduce the original results. The paper reports approx 70% mIoU, but the submitted paper reports 65.24. Since the code is not publicly accessible, the performance degradation may be possible if the authors reimplemented Yu et al.'s method.
However, since what the original paper reports are the values from the official leaderboard for the Waymo's competition. Readers consider the methods on the leaderboard to be fairly evaluated, and the reported accuracy is not in question (i.e., we don't need to reimplement the method) as we use the same experimental setting as the official competition. Also, we can see many methods outperform the proposed method in the official leaderboard.
https://waymo.com/open/challenges/2022/3d-semantic-segmentation/

Therefore, I cannot conclude that the proposed method achieves SOTA performance in the 3D semantic segmentation task. I understand that achieving SOTA accuracy is not the primary goal of academic study. However, the submitted paper argues a contribution that the proposed method achieves SOTA, which is invalid, and they should seek other ways to show their technical contribution.

**Questions:**

- Relation with instance/panoptic segmentation methods.
- Reasonable reasons for selecting the method by Yu et al. as the baseline, among other recent methods releasing implementations.

**Details Of Ethics Concerns:**

n.a

---

### Official Review · Reviewer_WFCE · 2023-11-03

**Soundness:** 3 good
**Presentation:** 3 good
**Contribution:** 2 fair
**Rating:** 5
**Confidence:** 4

**Summary:**

This paper proposes a new method for semantic segmentation of large 3D point clouds with deep neural networks, by using additional classification and reconstruction losses applied at the _instance_ level. These losses force the segments to better match the instances, and thereby improve the overall segmentation. The losses do not require additional supervision beyond per-point semantic labels, which are used to cluster the training point clouds into instances.

**Strengths:**

The idea of applying macro (instance level) losses to supervise point-wise estimates is a good one. The method appears to work well with a range of backbones and shows state-of-the-art results.

**Weaknesses:**

I am a bit confused about the instance extraction. I can come up with two interpretations:

1. The ground truth semantic labels are used by the clustering algorithm to segment the point clouds into object instances at the beginning of training. This is done once and not repeated. OR,

2. The predicted labels are used to recompute the predicted instances at each step of clustering. These predicted instances are compared to ground truth labels and instances for the losses.

I think the correct interpretation is (1). But just in case it is (2), how are gradients computed given the mean shift clustering step in the middle? I assume that every time the weights are updated during training, the predicted point labels change and the clusters need to be updated before the losses can be computed.

If indeed it is (1), then the focus of the method on large urban point clouds is clear: classical clustering algorithms work especially well when the clusters are well separated. The assertion that the method is "almost unsupervised" w.r.t. instance labels is a bit specious then, since this is a very favorable case for instance extraction. However, I am curious if the method could still work on more tight layouts like part assemblies of single shapes.

Apart from this question, I don't have significant methodological concerns. My main issue is with the scale of the contribution. Instance/part-level losses are not entirely new, e.g. old papers such as Kumar et al., "OBJ CUT", CVPR 2005 or newer ones like Zhu et al, "AdaCoSeg", CVPR 2020 or Sizikova et al. "Structure-Aware Shape Synthesis", 3DV 2018. While the general idea of applying instance-level losses is well-justified, I am not sure whether this is enough to push the paper over the acceptance bar given that the technical details (i.e. the classification/reconstruction heads), while reasonable, are not particularly surprising or insightful by themselves.

**Questions:**

See above.